# Indications, Detection, Completion and Retention Rates of Capsule Endoscopy in Two Decades of Use: A Systematic Review and Meta-Analysis

**DOI:** 10.3390/diagnostics12051105

**Published:** 2022-04-28

**Authors:** Pablo Cortegoso Valdivia, Karolina Skonieczna-Żydecka, Alfonso Elosua, Martina Sciberras, Stefania Piccirelli, Maria Rullan, Trevor Tabone, Katarzyna Gawel, Adam Stachowski, Artur Lemiński, Wojciech Marlicz, Ignacio Fernández-Urién, Pierre Ellul, Cristiano Spada, Marco Pennazio, Ervin Toth, Anastasios Koulaouzidis

**Affiliations:** 1Gastroenterology and Endoscopy Unit, University Hospital of Parma, University of Parma, 43126 Parma, Italy; 2Department of Biochemical Sciences, Pomeranian Medical University, 71-460 Szczecin, Poland; karzyd@pum.edu.pl; 3Gastroenterology Unit, Hospital García Orcoyen, 31200 Estella, Spain; alfonso.elosua@gmail.com (A.E.); maria_rullan@hotmail.com (M.R.); 4Division of Gastroenterology, Mater Dei Hospital, MSD 2080 Msida, Malta; muscatmartina@googlemail.com (M.S.); trev.tabone@gmail.com (T.T.); ellul.pierre@gmail.com (P.E.); 5Digestive Endoscopy Unit and Gastroenterology, Fondazione Poliambulanza, 25124 Brescia, Italy; stefania.piccirelli@poliambulanza.it (S.P.); cristianospada@gmail.com (C.S.); 6Fondazione Policlinico Universitario A. Gemelli IRCCS, Università Cattolica del Sacro Cuore, 00168 Rome, Italy; 7Department of Gastroenterology, Pomeranian Medical University, 70-001 Szczecin, Poland; ka.gawel88@gmail.com (K.G.); wojciech.marlicz@sanprobi.pl (W.M.); 8Department of Human Nutrition and Metabolomics, Pomeranian Medical University, 70-001 Szczecin, Poland; adam.stachowski@onet.pl; 9Department of Urology and Urological Oncology, Pomeranian Medical University, 70-001 Szczecin, Poland; artur.leminski@pum.edu.pl; 10Endoklinika, The Centre for Digestive Diseases, 70-535 Szczecin, Poland; 11Department of Gastroenterology, University Hospital of Navarra, 31008 Pamplona, Spain; ifurien@yahoo.es; 12Digestive Endoscopy Unit, Università Cattolica del Sacro Cuore, 00168 Rome, Italy; 13University Division of Gastroenterology, City of Health and Science University Hospital, University of Turin, 10126 Turin, Italy; pennazio.marco@gmail.com; 14Department of Gastroenterology, Skåne University Hospital, Lund University, 20502 Malmö, Sweden; ervin.toth@med.lu.se; 15Department of Medicine, OUH Svendborg Sygehus, 5700 Svendborg, Denmark; akoulaouzidis@hotmail.com; 16Department of Clinical Research, University of Southern Denmark (SDU), 5230 Odense, Denmark; 17Surgical Research Unit, OUH, 5000 Odense, Denmark; 18Department of Social Medicine and Public Health, Pomeranian Medical University, 70-001 Szczecin, Poland

**Keywords:** capsule endoscopy, systematic review, detection, indications, completion

## Abstract

Background: Capsule endoscopy (CE) has become a widespread modality for non-invasive evaluation of the gastrointestinal (GI) tract, with several CE models having been developed throughout the years. The aim of this systematic review and meta-analysis is to evaluate performance measures such as completion, detection and retention rates of CE. Methods: Literature through to August 2021 was screened for articles regarding all capsule types: small bowel, double-headed capsule for the colon or PillCam^®^Crohn’s capsule, magnetically-controlled capsule endoscopy, esophageal capsule and patency capsule. Primary outcomes included detection rate (DR), completion rate (CR) and capsule retention rate (RR). DR, CR and RR were also analyzed in relation to indications such as obscure GI bleeding (OGIB), known/suspected Crohn’s disease (CD), celiac disease (CeD), neoplastic lesions (NL) and clinical symptoms (CS). Results: 328 original articles involving 86,930 patients who underwent CE were included. OGIB was the most common indication (n = 44,750), followed by CS (n = 17,897), CD (n = 11,299), NL (n = 4989) and CeD (n = 947). The most used capsule type was small bowel CE in 236 studies. DR, CR and RR for all indications were 59%, 89.6% and 2%, respectively. According to specific indications: DR were 55%, 66%, 63%, 52% and 62%; CR were 90.6%, 86.5%, 78.2%, 94% and 92.8%; and RR were 2%, 4%, 1%, 6% and 2%. Conclusions: Pooled DR, CR and RR are acceptable for all capsule types. OGIB is the most common indication for CE. Technological advancements have expanded the scope of CE devices in detecting GI pathology with acceptable rates for a complete examination.

## 1. Introduction

Since its introduction into clinical practice more than two decades ago [1], wireless capsule endoscopy (CE) has become an indispensable diagnostic modality for the small bowel (SB) due to its non-invasive nature. As a result, its diagnostic role has been expanded to include, apart from the investigation of obscure gastrointestinal bleeding (OGIB), that of inflammatory bowel disease (IBD), polyposis syndromes and celiac disease (CeD), among others. An infrequent but potentially serious adverse event is capsule retention. Although retention can be managed conservatively in most cases, occasionally it requires endoscopic or surgical intervention. Published capsule retention rates (RR) vary depending on the background indication [2,3] and the use of a patency capsule (PC), a radiopaque dissolvable capsule with an equivalent size and shape as its electronic counterpart. PC use has proven safe and efficient to accurately assess SB functional patency [4].

Furthermore, technological breakthroughs prompted the development of additional CE models to non-invasively evaluate other segments of the gastrointestinal (GI) tract. For example, with the release of the colon capsule endoscopy (CCE) (2006) and the PillCam^®^Crohn’s capsule (PCC) (2017) (which allows a pan-enteric exploration in a single procedure [5]), CE again disrupted GI diagnostics. Moreover, recent studies are looking into magnetically controlled capsules (MCCE) for gastric evaluation [6] or combined gastric and SB assessment [7]. Therefore, we aimed to perform a systematic review and meta-analysis of the available literature concerning lesion detection, examination completion and capsule RR for all commercially available capsule models (i.e., esophageal, gastric/MCCE, SB, CCE (or pan-enteric) and PC), based on procedure indications.

## 2. Materials and Methods

### 2.1. Search Strategy and Inclusion Criteria

Four of the authors (M.S., S.P., T.T., K.G.) independently searched PubMed/MEDLINE/Embase/Ebsco/ClinicalTrials (from databases’ inception until 17 August 2021) for studies presenting CE detection, completion and/or retention rates. We included studies that provided data on CEs performed in adults only, with study groups comprising at least 30 participants. We excluded reviews/systematic reviews, editorials/perspectives/opinion pieces, individual case reports, letters to editors/commentaries and study protocols. The search strings we used for each database are available in Appendix B. The electronic search was followed by a manual review of the reference lists of relevant systematic reviews. The study was registered at the PROSPERO international register of systematic reviews (ID: 311560).

### 2.2. Data Abstraction 

We abstracted data on the study design, country, aims, patient groups (age, gender) and the type of capsule used according to the Preferred Reporting Items for Systematic Reviews and Meta-Analyses (PRISMA) standard [8]. Next, we looked for data on CE indications and grouped them as OGIB, Crohn’s disease (CD) (diagnostic workup or follow-up), neoplastic lesions (NL) or CeD. Other indications for CE that did not fit in any of the aforementioned groups are presented herein as clinical symptoms (CS). As the number of studies included in the final step exceeded 300, six independent investigators performed this step (A.E., M.S., S.P., M.R., T.T., K.G.). Capsule-type groups were defined as: capsule for the small bowel (SBCE), double-headed capsule for the colon (CCE) or PillCam^®^Crohn’s capsule (PCC), magnetically-controlled capsule endoscopy (MCCE), esophageal capsule (ESO) and patency capsule (PC). Whenever data were missing for the review, the authors of individual studies were contacted for additional information via email twice, two weeks apart. Consensus resolved any inconsistencies, with the last author/guarantor (A.K.) acting as adjudicator where necessary.

### 2.3. Outcomes

Primary outcomes were the rates of: (a) lesion detection (DR); (b) examination completion (CR); and (c) capsule retention. These were analyzed for specific indications and for all of the indications we evaluated. Outcomes were defined according to the definitions provided by the European Society of Gastrointestinal Endoscopy [9]; the definition of completion was based on the visualization of specific landmarks before the end of the recording: for SBCE procedures, imaging of the cecum; for CCE and PCC procedures, visualization of the anal verge/hemorrhoidal plexus; for ESO procedures, images of the stomach mucosa; for MCCE procedures, complete visualization of all anatomical gastric segments (i.e., cardia, fundus, body, incisura, antrum and pylorus). For PC procedures, completion was defined as either capsule excretion or radiological evidence of the capsule in the colon 30 h after ingestion.

### 2.4. Data Synthesis and Statistical Analysis

We conducted a random effects [10] model meta-analysis of outcomes when ≥3 studies contributed data using Comprehensive Meta-Analysis V3 (http://www.meta-analysis.com; last access 21 February 2022). Verification of abstracted data was performed by two separate investigators (A.S., A.L.). We explored study heterogeneity using the Chi-square test of homogeneity, with *p* < 0.05 indicating significant heterogeneity. All analyses were two-tailed with α = 0.05. The effect size that was measured was an event rate−in particular, DR, CR and RR. Subgroup analyses regarding the type of capsule were conducted. We conducted subgroup and exploratory maximum likelihood random effects meta-regression analyses of the co-primary outcomes, for all indication event rates. Meta-regression variables included:The year of publication (continuous moderator).The number of study participants (continuous moderator).The age of the participants (categorical moderator). We formulated the following ranges in the latter case: <60, 60–80 and >80 years old. Finally, we inspected funnel plots and used Egger’s regression test [11] and the Duval and Tweedie’s trim and fill method where applicable [12] to quantify whether publication bias could have influenced the results.

## 3. Results

### 3.1. Search Results

The initial search yielded 3241 hits. Two thousand seven hundred thirty-five (n = 2735) studies were excluded after identification as duplicates and/or after evaluation on the title/abstract level. Subsequently, we did not identify other studies via hand search. Eventually, 506 full-text articles were reviewed. Of those, 178 were excluded due to not fitting our inclusion criteria. Reasons for exclusion were: type of study (case report, review, letter) (n = 28); too few participants (n = 30); age of the subjects included in the study (n = 11); animal studies (n = 2); language other than English/Spanish/French/Greek/Polish/Italian (n = 11); not enough data available (n = 30); full text not available (n = 35); technique presentation/diagnostic algorithms with no direct clinical CE involvement (n = 31). Therefore, 328 studies were found eligible and included in this meta-analysis (Appendix A).

### 3.2. Study and Studied Subjects Characteristics

Altogether, 328 studies comprising 86,930 patients who underwent CEs were included in the final synthesis. We abstracted data from 122 retrospective and 206 prospective studies. Patients of both genders were included, with the highest reported percentage of males being 94.2%. The youngest mean age was 26, while the highest median age was 72. The most prevalent indication for CE was OGIB (n = 44,750), followed by CD (n = 11,299), NL (n = 4989) and CeD (n = 947). Unspecified CS was reported in 17,897 individuals. The most used capsule type was SBCE in 236 studies (Figure 1). Data are available in Appendix A.

### 3.3. Lesion Detection Rates (DR) by Capsule Type

The DR was calculated per indication group as a pooled event rate. This was either provided by the authors (with no information regarding particular lesion types) or calculated by ourselves (as the sum of the detected lesions per indication group). In addition, we conducted a comparative analysis by capsule type. The DR for all lesions (pooled data) differed significantly by capsule type, with the highest rate for PCC (DR = 0.693), followed by CCE (DR = 0.643). However, in the indication subgroup analyses, there were no significant differences in DRs by capsule type; data is presented in Table 1. Raw data can be found in Appendix A. Exemplary forest plots (indication subgroup—OGIB and CD) are depicted in Figure 2 and Appendix A.

Regarding DR, for all lesions, we conducted a meta-regression and found that neither the year of publication (coefficient = −0.019; standard error (SE) = 0.011, Z = −1.77, *p* = 0.076) nor the number of participants per study (coefficient = −0.0001; SE = 0.0001, Z = −1.23, *p* = 0.2196) or the age range (“<60”: coefficient = 0.067; SE = 0.1850, Z = 0.36, *p* = 0.7158; “60–80”: coefficient = 0.1617; SE = 0.1975, Z = 0.82, *p* = 0.4129) influenced study-level effect sizes. We also inspected funnel plots and found that Egger’s test did not suggest a publication bias regarding the net DR for all indications (OGIB: *p* = 0.612; CD: *p* = 0.111; NL: *p* = 0.232; CeD: *p* = 0.155), except for DRs in CS (*p* = 0.029) and for all lesions (*p* = 0.040). In the former, the Duval and Tweedie method adjusted values of 13 studies to the left of the mean; random model point estimate 0.481; 95%CI 0.408–0.555, Q value = 1922.019; whilst in the latter, the approach-adjusted values of 44 studies to the left of the mean; random model point estimate 0.515; 95%CI 0.491–0.539, Q value = 8811.99.

### 3.4. Completion Rates (CR) by Capsule Type

The CRs were also calculated per indication group as pooled event rates. The OGIB and CD subgroup analyses by capsule type demonstrated no significant differences in CRs. However, the CRs for NL, CS and pooled data differed significantly by capsule type, with the highest rates for CCE (NL: CR = 0.921; Appendix A) and MCCE (CS: CR = 0.997; All indications: CR = 0.959). Data is presented in Table 2 and Appendix A. In the case of CRs for all lesions, we conducted a meta-regression and no association between the year of publication (coefficient = 0.000; SE = 0.013, Z = 0.00, *p* = 0.99), the number of patients (coefficient = 0.0001; SE = 0.0001, Z = 0.49, *p* = 0.6252), the age range (“<60”: coefficient = −0.0247; SE = 0.218, Z = −0.11, *p* = 0.91; “60–80”: coefficient = −0.078; SE = 0.239, Z = −0.33, *p* = 0.704) or the effect size was found. 

Egger’s test did suggest a publication bias regarding the net CR for all indications (OGIB: *p* = 0.00002; CD: *p* = 0.00001; CS: *p* = 0.00005; pooled indications *p* = 0.0000), apart from NL (*p* = 0.062) and CeD (*p* = 0.287) CRs. The Duval and Tweedie method-adjusted values were as follows: OGIB 13 studies to the left of the mean; random model point estimate: 0.868; 95%CI: 0.841–0.891, Q value = 609.240; CD: 11 studies to the left of the mean; random model point estimate: 0.785; 95%CI: 0.729–0.832, Q value = 295.78; CS: 13 studies to the left of the mean; random model point estimate: 0.857; 95%CI: 0.811–0.894, Q value = 348.054; pooled indications 61 studies to the left of the mean; random model point estimate: 0.836; 95%CI: 0.819–0.852, Q value = 5780.222.

### 3.5. Retention Rates (RR) by Capsule Type

The RRs did not differ significantly by capsule types in the OGIB and CD indication groups. There were, however, significant differences for other indications. In the case of NL and CS, the lowest RRs were for PC (NL: RR = 0.002; CS: RR = 0.002; Appendix A), whilst for pooled indications, the lowest RRs were found for CCE (RR = 0.008) and MCCE (RR = 0.01). Data is presented in Table 3 and Appendix A. RRs for all lesions were not influenced by any of the covariates (Year of publication: coefficient = −0.031; SE = 0.021, Z = −1.45, *p* = 0.146; Number of patients: coefficient = −0.0001; SE = 0.0001, Z = −1.24, *p* = 0.216; age range: “<60”: coefficient = 0.448; SE = 1.26, Z = 1.26, *p* = 0.206; “60–80”: coefficient = 0.123; SE = 0.387, Z = 0.32, *p* = 0.751). 

Egger’s test did suggest a publication bias regarding the net RR for almost all indications; OGIB: *p* = 0.046; CD: *p* = 0.019; NL: *p* = 0.00000; CS: *p* = 0.010; pooled lesions *p* = 0.0000). Only in the case of CeD was there no publication bias detected (*p* = 0.971). The Duval and Tweedie method-adjusted values were as follows: OGIB 20 studies to the right of the mean; random model point estimate: 0.022; 95%CI: 0.016–0.026, Q value = 257.505; CD: 12 studies to the right of the mean; random model point estimate: 0.064; 95%CI: 0.043–0.093, Q value = 579.807; NL: 8 studies to the right of the mean; random model point estimate: 0.023; 95%CI: 0.012–0.045, Q value = 93.036; CS: 18 studies to the right of the mean; random model point estimate: 0.046; 95%CI: 0.030–0.071, Q value = 473.375; pooled lesions 97 studies to the right of the mean; random model point estimate: 0.027; 95%CI: 0.023–0.032, Q value = 2104.689.

## 4. Discussion

This meta-analysis collected articles describing CE procedures in the last 20 years, focusing on specific performance indicators such as completion, detection and retention rates (Table 4). In 2010, Liao and colleagues [13] performed a systematic review with similar aims; however, they only extracted and analyzed SBCE outcomes because at that time, other CE literature was inevitably scarcer. Nevertheless, their work highlighted the acceptable SBCE safety profile and satisfactory pooled detection and CR of the analyzed studies. As technological advancements have allowed newer CE tools to enter the market, we decided that the time was ripe to provide a necessary, all-inclusive 2022 update by assimilating additional evidence in our analysis. Although several other monothematic meta-analyses have been published to date, including a detailed meta-analysis on adverse events for all types of CE by the same group [14]. This work provides a detailed overview of the entire spectrum of CE use in clinical practice.

Our analysis showed no significant difference in the pooled lesion detection per indication group in CE, although the highest DRs were seen with PCC and CCE. This overarching result supports published studies that advocate for routine use of double-headed CE in SB assessment, largely due to its enhanced detection potential and ability to change clinical diagnosis and patient management [15]. Naturally, CD and NL were the indication group with the highest DR, with percentages of 66% and 63%, respectively. One explanation for this could be a more selective allocation in these patient groups and the expected higher incidence of colonic neoplasia when compared to SB NL.

Overall, completion was obtained in 89.6% of the procedures (all types of CE), in line with previous results in the literature [13]. OGIB and CD, the most represented indications, did not show any statistically significant difference in CR (90.6% and 86.5%, respectively), or even in the subgroup analyses per capsule type. CCE was used in the investigation of colonic NL in 11 studies, with a satisfactory CR of over 92%. Retention, probably the most cumbersome adverse event in CE, is an uncommon complication (<1% of the procedures), which is known to be reduced by the use of PC [14] and favored by underlying CD [3]. In our study, the overall RR was as low as 2%, with the highest values of 4% and 6% in CD and CeD, respectively. The highest values seen in CeD can likely be explained by heterogeneity due to the low number of studies (only 6) compared to other indications (68, 52, 50 and 23, respectively, for OGIB, CS, CD and NL).

This study has a number of limitations. First, the heterogeneity of the included studies in regard to the terminology of measured outcomes and data presentation, with publication bias (shown by Egger’s tests) on outcomes of CRs and RRs. Secondly, the exclusion of studies with <30 participants may add uniformity to the results while excluding potentially useful data from the analysis. Last, the meta-regression was performed at a study level, thereby not allowing further analyses on the demographical data.

## 5. Conclusions

In conclusion, in the last 20 years, CE has confirmed its substantial role in GI examination. With an excellent safety profile, technological advancements have expanded the scope of CE devices in the detection of GI pathology with acceptable rates for complete examination; however, pitfalls still persist (e.g., capsule retention in CD patients, optimal and shared bowel preparation regimens). It is expected that the widespread adoption of AI-based technologies, which provide high profiles of pathology detection and characterization, will further enhance the performance outcomes of CE [16].

## Figures and Tables

**Figure 1 diagnostics-12-01105-f001:**
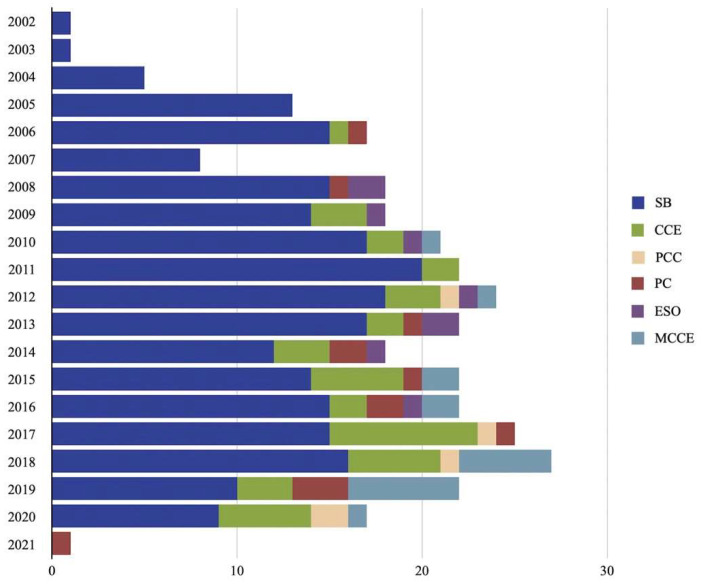
Yearly publication of included studies per type of capsule.

**Figure 2 diagnostics-12-01105-f002:**
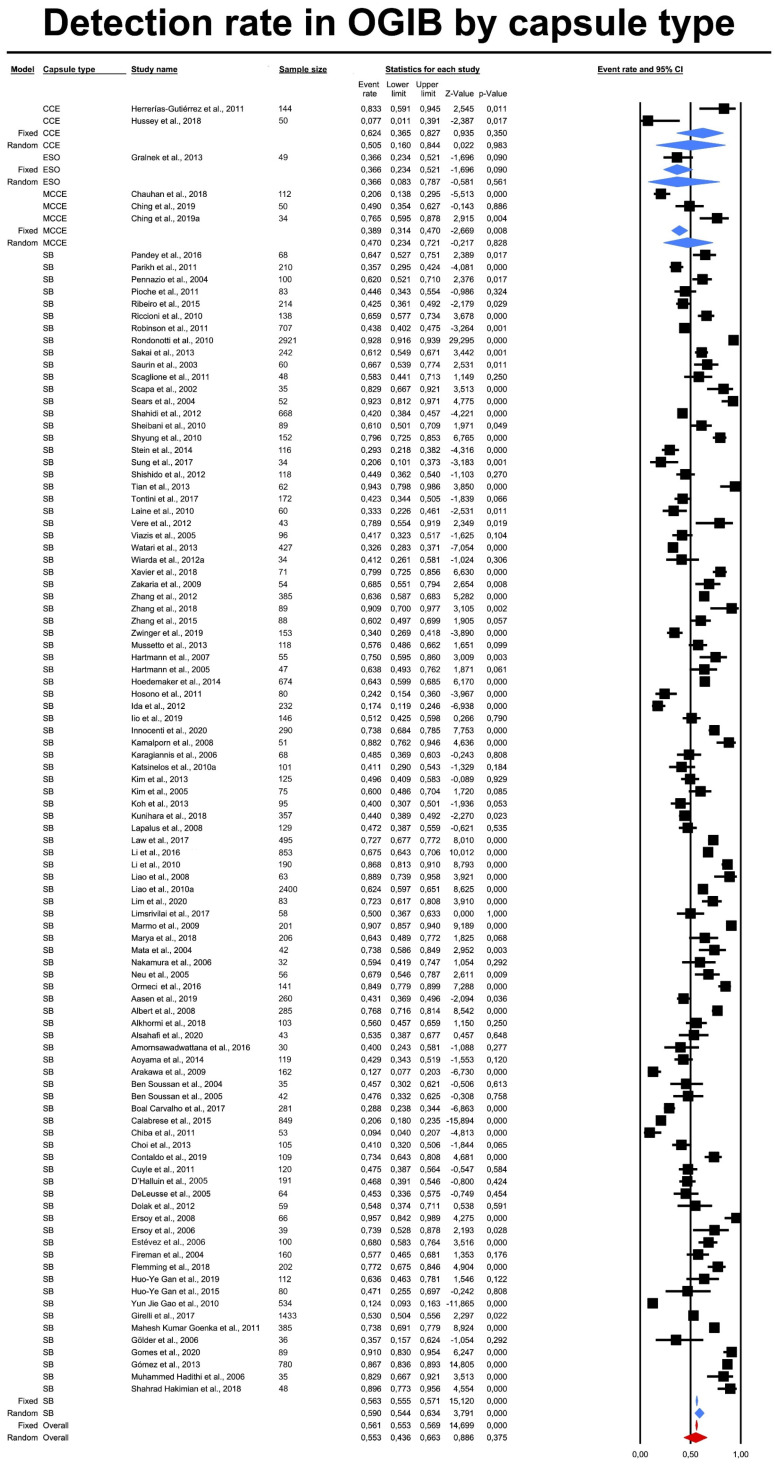
Detection rates in obscure gastrointestinal bleeding by capsule type.

**Table 1 diagnostics-12-01105-t001:** Detection rates by type of capsule endoscope.

Capsule Type	Effect Size and 95%CI				Test Z		Heterogenity (from Fixed Effect Analysis)			
	Number of Studies	Point Estimate	Lower Limit	Upper Limit	z Value	*p* Value	Q Value	df(Q)	*p* Value	I^2^
**OGIB**										
CCE	2	0.50	0.16	0.84	0.02	0.98	11.30	1.00	0.00	91.15
ESO	1	0.37	0.08	0.79	−0.58	0.56	0.00	0.00	1.00	0.00
MCCE	3	0.47	0.23	0.72	−0.22	0.83	31.78	2.00	0.00	93.71
SBCE	95	0.59	0.54	0.63	3.79	0.00	2814.10	94.00	0.00	96.66
Total between							1.79	3.00	0.62	
Overall	101	0.55	0.44	0.66	0.89	0.38	2880.59	100.00	0.00	96.53
**CD**										
CCE	2	0.82	0.60	0.93	2.66	0.01	2.37	1.00	0.12	57.86
Combi	1	0.52	0.19	0.83	0.09	0.93	0.00	0.00	1.00	0.00
PCC	4	0.68	0.48	0.83	1.74	0.08	12.94	3.00	0.00	76.81
SBCE	36	0.62	0.55	0.68	3.42	0.00	301.31	35.00	0.00	88.38
Total between							3.71	3.00	0.29	
Overall	43	0.66	0.53	0.77	2.38	0.02	345.07	42.00	0.00	87.83
**NL**										
CCE	12	0.67	0.55	0.78	2.73	0.01	366.85	11.00	0.00	97.00
SBCE	7	0.56	0.38	0.72	0.62	0.53	104.46	6.00	0.00	94.26
Total between							1.19	1.00	0.27	
Overall	19	0.63	0.52	0.73	2.27	0.02	478.65	18.00	0.00	96.24
**CeD**										
SBCE	9	0.52	0.40	0.64	0.37	0.71	39.63	8.00	0.00	79.81
Total between							0.00	0.00	1.00	
Overall	9	0.52	0.40	0.64	0.37	0.71	39.63	8.00	0.00	79.81
**CS**										
CCE	11	0.60	0.42	0.75	1.09	0.28	155.06	10.00	0.00	93.55
ESO	7	0.68	0.48	0.84	1.74	0.08	113.49	6.00	0.00	94.71
MCCE	4	0.68	0.40	0.87	1.29	0.20	107.81	3.00	0.00	97.22
PCC	2	0.84	0.43	0.97	1.66	0.10	2.32	1.00	0.13	56.93
SBCE	41	0.55	0.46	0.64	1.02	0.31	907.32	40.00	0.00	95.59
Total between							3.82	4.00	0.43	
Overall	65	0.62	0.51	0.73	2.07	0.04	1371.25	64.00	0.00	95.33
**All indications**										
CCE	38	0.64	0.58	0.70	4.71	0.00	738.68	37.00	0.00	94.99
Combi	2	0.58	0.34	0.79	0.65	0.51	3.39	1.00	0.07	70.46
ESO	9	0.59	0.46	0.70	1.39	0.16	160.41	8.00	0.00	95.01
MCCE	17	0.47	0.38	0.56	−0.61	0.54	616.48	16.00	0.00	97.40
PCC	5	0.69	0.53	0.82	2.24	0.03	17.71	4.00	0.00	77.42
SBCE	202	0.57	0.55	0.60	5.49	0.00	4316.24	201.00	0.00	95.34
Total between							11.92	5.00	0.04	
Overall	273	0.59	0.52	0.65	2.65	0.01	7360.86	272.00	0.00	96.30

Abbreviations: CCE: colon capsule endoscopy; CD: Crohn’s disease; CeD; celiac disease; CI: confidence interval; Combi: different types of capsules; CS: clinical symptoms; ESO: esophageal capsule; MCCE: magnetically controlled capsule endoscopy; NL: neoplastic lesions; OGIB: obscure gastrointestinal bleeding; PCC: PillCam^®^Crohn’s capsule; SBCE: small bowel capsule endoscopy.

**Table 2 diagnostics-12-01105-t002:** Completion rates by type of capsule endoscope.

Capsule Type	Effect Size and 95%CI				Test Z		Heterogenity (from Fixed Effect Analysis)			
	Number of Studies	Point Estimate	Lower Limit	Upper Limit	z Value	*p* Value	Q Value	df(Q)	*p* Value	I^2^
**OGIB**										
CCE	1	0.742	0.398	0.926	1.407	0.159	0.000	0.000	1.000	0.000
ESO	1	0.978	0.803	0.998	3.107	0.002	0.000	0.000	1.000	0.000
MCCE	1	0.978	0.667	0.999	2.396	0.017	0.000	0.000	1.000	0.000
SBCE	56	0.891	0.868	0.910	1.894	0.000	508.477	55.000	0.000	89.183
Total between							5.031	3.000	0.170	
Overall	59	0.906	0.753	0.968	3848	0.000	519.524	58.000	0.000	88.836
**CD**										
CCE	2	0.702	0.632	0.764	5.300	1.16 × 10^−7^	4.517604	1.000	0.034	77.864
Combi	1	0.991	0.875	0.999	3.328	8.74 × 10^−4^	6.67 × 10^−14^	0.000	1.000	0.000
PC	2	0.893	0.827	0.935	7.480	7.46 × 10^−14^	0.195398	1.000	0.658	0.000
PCC	4	0.879	0.828	0.916	9.521	0	3.270906	3.000	0.352	8.282
SBCE	24	0.717	0.696	0.737	18.231	0	0.0187	23.000	0.000	87.704
Total between							6.957	4.000	0.138	
Overall	33	0.865	0.766	0.927	5.410	6.30 × 10^−8^	242.297	32.000	0.000	86.793
**NL**										
CCE	11	0.921	0.860	0.957	7.467	0.000	155.2934	10.000	0.000	93.561
PC	1	0.496	0.127	0.870	−0.015	0.988	2.07 × 10^−17^	0.000	1.000	0.000
SBCE	6	0.707	0.512	0.848	2.073	0.038	56.14724	5.000	0.000	91.095
Total between							12.024	2.000	0.002	
Overall	18	0.782	0.455	0.939	1.718	0.086	409.476	17.000	0.000	95.84835
**CeD**										
SBCE	5	0.940	0.836	0.980	4.817	1.46 × 10^−6^	19.41264	4.000	0.001	79.395
Overall	5	0.940	0.836	0.980	4.817	1.46 × 10^−6^	19.41264	4.000	0.001	79.395
**CS**										
CCE	8	0.888	0.790	0.944	5.442	0.000	57.4281	7.000	0.000	87.811
ESO	4	0.817	0.642	0.918	3.211	0.001	50.21082	3.000	0.000	94.025
MCCE	3	0.997	0.982	1.000	6.225	0.000	9.61 × 10^−3^	2.000	0.995	0.000
PC	1	0.870	0.562	0.972	2.257	0.024	0	0.000	1.000	0.000
PCC	2	0.960	0.831	0.992	3.911	0.000	7.12 × 10^−4^	1.000	0..979	0.000
SBCE	23	0.902	0.859	0.932	10.607	0.000	90.13725	22.000	0.000	75.593
Total between							18.916	5.000	0.002	
Overall	41	0.928	0.845	0.968	5.847	5.01 × 10^−9^	277.4474	40.000	0.000	85.583
**All indications**										
CCE	42	0.857	0.818	0.889	12.255	0.000	485.6277	41.000	0.000	91.557
Combi	3	0.953	0.872	0.984	5.390	0.000	467.6693	2.000	0.000	99.572
ESO	5	0.859	0.712	0.938	3.936	0.000	58.78379	4.000	0.000	93.195
MCCE	13	0.959	0.924	0.978	9.615	0.000	389.2855	12.000	0.000	96.917
PC	12	0.846	0.764	0.903	6.333	0.000	397.1725	11.000	0.000	97.230
PCC	5	0.920	0.825	0.965	5.399	0.000	8.72	4.000	0.069	54.123
SBCE	177	0.876	0.860	0.890	27.449	0.000	2356.912	176.000	0.000	92.533
Total between							20.125	6.000	0.003	
Overall	257	0.896	0.857	0.925	11.640	0.000	4681.25	256.000	0.000	94.531

Abbreviations: CCE: colon capsule endoscopy; CD: Crohn’s disease; CeD: celiac disease; CI: confidence interval; Combi: different type of capsules; CS: clinical symptoms; ESO: esophageal capsule; MCCE: magnetically controlled capsule endoscopy; NL: neoplastic lesions; OGIB: obscure gastrointestinal bleeding; PC: patency capsule; PCC: PillCam^®^Crohn’s capsule; SBCE: small bowel capsule endoscopy.

**Table 3 diagnostics-12-01105-t003:** Retention rates by type of capsule endoscope.

Capsule Type	Effect Size and 95%CI				Test Z		Heterogenity (from Fixed Effect Analysis)			
	Number of Studies	Point Estimate	Lower Limit	Upper Limit	z Value	*p* Value	Q Value	df(Q)	*p* Value	I^2^
**OGIB**										
CCE	3	0.04	0.01	0.21	−3.39	0.00	1.48	2.00	0.48	0.00
Combi	2	0.01	0.00	0.03	−9.10	0.00	45.28	1.00	0.00	97.79
MCCE	2	0.06	0.01	0.37	−2.42	0.02	1.09	1.00	0.30	7.94
PC	1	0.02	0.00	0.12	−3.98	0.00	0.00	0.00	1.00	0.00
SBCE	60	0.01	0.01	0.02	−30.98	0.00	102.89	59.00	0.00	42.66
Total between							3.22	4.00	0.52	
Overall	68	0.02	0.01	0.03	−13.02	0.00	157.76	67.00	0.00	57.53
**CD**										
CCE	1	0.01	0.00	0.16	−2.95	0.00	0.00	0.00	1.00	0.00
Combi	2	0.04	0.01	0.22	−3.27	0.00	5.08	1.00	0.02	80.33
MCCE	1	0.11	0.01	0.71	−1.37	0.17	0.00	0.00	1.00	0/00
PC	4	0.08	0.02	0.26	−3.42	0.00	34.61	3.00	0.00	91.33
PCC	3	0.03	0.01	0.17	−3.64	0.00	3.94	2.00	0.14	49.29
SBCE	39	0.04	0.03	0.07	−11.86	0.00	390.55	38.00	0.00	90.27
Total between							2.46	5.00	0.78	
Overall	50	0.04	0.02	0.09	−8.67	0.00	520.57	49.00	0.00	90.59
**NL**										
CCE	11	0.00	0.00	0.01	−12.08	0.00	9.58	10.00	0.48	0.00
Combi	1	0.01	0.00	0.06	−4.40	0.00	0.00	0.00	1.00	0.00
MCCE	1	0.01	0.00	0.19	−2.91	0.00	0.00	0.00	1.00	0.00
PC	1	0.00	0.00	0.04	−4.11	0.00	0.00	0.00	1.00	0.00
SBCE	9	0.04	0.02	0.07	−9.64	0.00	16.55	8.00	0.04	51.65
Total between							21.23	4.00	0.00	
Overall	23	0.01	0.00	0.04	−5.82	0.00	69.32	22.00	0.00	68.26
**CeD**										
SBCE	6	0.06	0.00	0.48	−2.01	0.04	50.01	5.00	0.00	90.00
Overall	6	0.06	0.00	0.48	−2.01	0.04	50.01	5.00	0.00	90.00
**CS**										
CCE	7	0.02	0.00	0.06	−5.79	0.00	1.77	6.00	0.94	0.00
Combi	2	0.04	0.01	0.22	−3.30	0.00	27.51	1.00	0.00	96.36
ESO	3	0.25	0.05	0.69	−1.13	0.26	51.67	2.00	0.00	96.13
MCCE	5	0.01	0.00	0.03	−6.07	0.00	1.81	4.00	0.77	0.00
PC	1	0.00	0.00	0.07	−3.31	0.00	0.00	0.00	1.00	0.00
PCC	1	0.01	0.00	0.21	−2.78	0.01	0.00	0.00	1.00	0.00
SBCE	33	0.03	0.01	0.04	−12.51	0.00	157.35	32.00	0.00	79.66
Total between							12.96	6.00	0.04	
Overall	52	0.02	0.01	0.06	−6.80	0.00	426.64	51.00	0.00	88.05
**All indications**										
CCE	42	0.01	0.00	0.01	−17.99	0.00	156.98	41.00	0.00	73.88
Combi	3	0.02	0.00	0.08	−5.39	0.00	5.92	2.00	0.05	66.22
ESO	9	0.04	0.01	0.11	−5.62	0.00	146.12	8.00	0.00	94.52
MCCE	17	0.01	0.00	0.02	−11.69	0.00	49.67	16.00	0.00	67.79
PC	12	0.05	0.03	0.10	−7.76	0.00	145.24	11.00	0.00	92.43
PCC	5	0.02	0.01	0.08	−5.62	0.00	7.29	4.00	0.12	45.16
SBCE	184	0.02	0.01	0.02	−37.76	0.00	937.46	183.00	0.00	80.48
Total between							21.50	6.00	0.00	
Overall	272	0.02	0.01	0.03	−14.18	0.00	1832.05	271.00	0.00	85.21

Abbreviations: CCE: colon capsule endoscopy; CD: Crohn’s disease; CeD: celiac disease; CI: confidence interval; Combi: different type of capsules; CS: clinical symptoms; ESO: esophageal capsule; MCCE: magnetically controlled capsule endoscopy; NL: neoplastic lesions; OGIB: obscure gastrointestinal bleeding; PC: patency capsule; PCC: PillCam^®^Crohn’s capsule; SBCE: small bowel capsule endoscopy.

**Table 4 diagnostics-12-01105-t004:** Detection, completion and retention rate of capsule endoscopy (all types) according to indications.

Indications	Detection Rate	Completion Rate	Retention Rate
OGIB	55%	90.6%	2%
CD	66%	86.5%	4%
NL	63%	78.2%	1%
CeD	52%	94.0%	6%
CS	62%	92.8%	2%
All Indications	59%	89.6%	2%

Abbreviations: CD: Crohn’s disease; CeD: celiac disease; CS: clinical symptoms; NL: neoplastic lesions; OGIB: obscure gastrointestinal bleeding.

## Data Availability

Not applicable.

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
