# Peer review of "Indications, Detection, Completion and Retention Rates of Capsule Endoscopy in Two Decades of Use: A Systematic Review and Meta-Analysis"

_diagnostics, 2022, doi:10.3390/diagnostics12051105_

Round 1

Reviewer 1 Report

Thank you for your submission. Please find my questions below.

Q1. ln 135 age subjects included in the study (n= 11)
    were these 11 excluded because they contained pediatric patients?

Q2. If possible, could you please add sample size to figure 2? The sample size in the plot would help the reader look at the influence (or weight) of each paper.

Q3. Did you consider using study type as a variable in the meta-regression 
as well? If so, was there a significant difference in effect-size between the two, retrospective and prospective?

Thank you.

Author Response

Q1: Yes. As specified in the inclusion criteria, only studies involving adult patients were included in the analysis – therefore, study regarding pediatric patients were excluded

Q2: Figure 2 has been edited according to your suggestion, now showing the sample size of each study

Q3: The outcomes that were considered for meta-regression were 1) year of publication 2) number of study participants 3) age of the participants. We discussed the role of adding other possible variables before performing this statistical analysis, but we decided to maintain only those three due to the high heterogeneity of included studies; this heterogeneity, which also represents a major limitation of our study, is due not only to the various type of studies but also to the variability of outcomes that the authors considered and the long time frame of production. According to these reasons, we did not believe that this outcome could add any relevant difference in the meta-regression results. We hope this explanation can answer your question properly.

Reviewer 2 Report

Authors present a systematic review and meta-analysis of 328 studies on the efficacy and safety profile of several brands of commercial capsule endoscopy devices in a range of GI disorders.

  1. Authors should provide the specific definition of key outcome measures “detection rate”, “completion” and “retention”. The following questions should be address in the definition: 
    1. Does detection rate mean positive diagnosis of disorder or identification of possible abnormalities? Are diagnosis based on capsule alone or in combination with other diagnostic tools?
    2. Does an incomplete study mean technical malfunction or human error during data recording?
    3. Does retention mean capsule retained in GI due to running out of battery? What were the consequence of retention (e.g. laxative use, endoscopy, surgery)?

The manuscript provides sufficient review of relevant literature and data on the efficacy and safety profiles. However, key outcome measures are not defined. Leaving the readers confused on the consequences of potential risks (for example, use of capsules endoscopies in celiac disease appear to have significant risk (6%) but low benefit (52% detection), what happens in the 6% that have retention and 48% that failed detection)?

The authors fail to discuss the possible reasoning’s and clinical consequences behind the differences in detection, completion, retention rate between capsule and indications. Despite the comprehensive and thorough review of literature and data analysis, this manuscript lacks relevance and applicable information for clinicians to make an informed choice.

Line 262 – “Retention rates as low as 0.02%, with the highest values of 0.04% and 0.06%” but in Table 4 retention rates are between 1 – 6%.

Author Response

Thank you for the comments you provided.

Q1: As you underline, detection, completion and retention rates were used as key outcomes of our study. Since the definition of these outcomes is widely known to readers approaching this type of manuscript, we sticked to the ones proposed by ESGE (stated with reference n.9 in the methods). Nevertheless, we provided a more precise definition of “completion” since different types of capsules were taken in consideration.

  • Detection rate in CE is considered whenever a possible abnormality / disorder is encountered during capsule examination. Hereby, we sticked to the definition of “detection” and not “diagnosis” (which by all means may not be always provided with CE alone)
  • A study is considered as incomplete whenever the capsule fails to show predefined landmarks (variable according to the capsule type); the causes can be multiple: battery exhaustion, physical obstacles slowing down the capsule transit (stenoses, masses, delayed gastric emptying…), technical problems, poor bowel prep.
  • Retention is considered whenever the capsule is not excreted naturally from the GI tract after 2 weeks from ingestion. Consequences of retention may vary, according to the cause of retention – in the present study, we did not extracted data on strategies after CE retention.

As explained in the discussion, the high values for retention in celiac disease may probably be explained by heterogeneity of data due to the low number of studies (only 6) compared to other indications (68, 52, 50 and 23 respectively for OGIB, CS, CD and NL). Regarding the 48% of patients failing detection in the same setting, it means that in these patients no endoscopic abnormalities were seen by CE – according to the guidelines, CE may have a role in seronegative celiac disease or whenever the patient shows abnormal serum biomarkers but no atrophy is detected by the biopsies. Therefore, the explanation may be that in this subset of patients no other lesions were highlighted by CE.

Regarding line 262, 2% is the OVERALL retention rate, whereas 4 and 6% are the highest values (Crohn’s and celiac disease). The line has been edited accordingly, as the values in decimal are the point estimate – the correct percentages are 2, 4 and 6.
